# Age-specific trends in limitations of daily activities in American adults aged 50–84 by race and ethnicity, 2000–2018

Octavio Bramajo[1]*, Brandon O'Grady[2], Moumita Chakraborty[3], Neil K. Mehta[4]

1 Anthropometrics and Historical Epidemiology Research Group, Institute of Evolutionary Medicine, University of Zurich, Switzerland; Sealy Center on Aging, University of Texas Medical Branch, Galveston, Texas, United States of America, 2 School of Health Professions, University of Texas Health Science Center at San Antonio, Galveston, Texas, United States of America, 3 Department of Biostatistics and Data Sciences, School of Public and Population Health, University of Texas Medical Branch, Galveston, Texas, United States of America, 4 Department of Epidemiology, School of Public and Population Health, University of Texas Medical Branch, Galveston, Texas, United States of America

* octavio.bramajo@iem.uzh.ch

## Abstract

During the first decade of the 2000s, disability levels began to stagnate or increase among American adults, a sharp departure from broad declines observed in earlier decades. It is unclear whether these adverse trends have continued after 2010, and whether there are variations by age group, sex, race/ethnicity, and nativity status. We used data from the 2000–2018 National Health Interview Survey to investigate trends in limitations of activities of daily living (ADL) and instrumental activities of daily living (IADL) by age group, race/ethnicity, and nativity among the 50+ population. We evaluated trends within the 2000–2009 and 2010–2018 periods. We find that adverse trends in disability documented in 2000–2009 continued into the 2010–2018 period, although the groups experiencing the largest increases shifted between decades. Middle-aged (50–64 years) adults displayed the most adverse trends compared to those ages 65–74 and 75–84, with some subgroups experiencing over 100% relative increases in ADL limitations. Disparities in disability by race/ethnicity widened considerably, with an alarming increase in disability among US-born Hispanics, particularly women at middle age, in the 2010–2018 period, while in the previous decade, Non-Hispanic Blacks and Non-Hispanic Whites presented the largest increases in IADL and ADL prevalence. Foreign-born Hispanics, who maintained the lowest disability levels through 2010, also experienced significant increases in the second decade. These trends provide an evidence base for understanding the direction of longer-term disability trajectories in the U.S. population and highlight the need for targeted prevention efforts among middle-aged adults and vulnerable demographic groups.

**Data availability statement:** All relevant data for this study are publicly available from the OSF repository (https://osf.io/6pfth).

**Funding:** The National Institute on Aging (R01AG075208, PI: NKM) provided funding for this investigation, but did not have any role in the design, conduct, and reporting of this study. The contents of this article are the sole responsibility of the authors.

**Competing interests:** The authors have declared that no competing interests exist.

## Introduction

**Improvements in life expectancy during the early 2000s gave way to a period of stagnation that has since become one of the most closely followed developments in population health** [1–6]. However, trends in disability among American adults over recent periods, have not been followed as closely.

Limitations in activities of daily living (ADLs) and instrumental activities of daily living (IADLs) have long been considered gold standard measures for assessing functional disability in population-based research [7–9]. These measures capture fundamental aspects of independent functioning—from basic self-care tasks like bathing and dressing (ADLs) to more complex activities like managing finances and medications (IADLs). These measures are often used in the context of the disablement process [10], a framework that demonstrates how chronic conditions, injuries, and impairments lead to functional limitations that hinder quality of life and the ability to perform daily activities [11,12]. Over the 1980s and 1990s, the U.S. experienced sizeable declines in national-level disability prevalence as measured by ADLs and IADLs [13–16], with declines ranging 1% and 2.5% per year since the 1990s [17]. Various explanations for this improvement have been identified including increases in educational attainment (Freedman et al., 2002; Martin et al., 2010; Schoeni et al., 2001), improvements in medical care, and the long-term benefits of improvements in earlier-in-life health [18,19]. The first decade of the 2000s brought about a notable shift in U.S. disability trends measured by these indicators. Previous studies that analyzed this period found that the declining ADL and IADL prevalence among Americans aged 55 and older stagnated [20,21,22,23]. Some other studies suggest that disability increased among middle-aged individuals, reaching by 2010 a 1.7% prevalence of ADL and 4% of IADL for those aged between 40 and 65, and above near 9% and 12% respectively for individuals aged 65+ [20,24,25]. Other group classifications presented similar trends: among those aged 55–69, IADL limitations increased by an average of 1.33% annually between 1998–2012, with most of the increase concentrated toward the end of the period [26].

Trends after 2010 remain understudied. Around 2010, U.S. life expectancy began to stagnate and even decline driven by stagnating declines in cardiovascular disease mortality and rising mortality from drug overdoses, suicides, and alcoholic liver disease. Broader deteriorations in population health have also been documented after 2010 [2,3]. The 2008 economic crisis and its aftermath provide additional motivation for examining the post-2010 period. Economic recessions have been shown to have lasting effects on population health, particularly through increased financial strain, job loss, reduced access to healthcare, and heightened psychological distress [27,28]. It remains unclear whether this event is also associated with a turning point for disability trends. These effects may be particularly pronounced for middle-aged adults approaching retirement who experienced job displacement during peak earning years. Moreover, the post-2010 period saw accelerating increases in obesity levels and the continued escalation of the opioid epidemic, which are key risk factors for functional limitations [29–31].

Investigating disability trends by race/ethnicity is also critical for a complete understanding of the demography of U.S. disability with wider implications for health and economic disparities. Non-Hispanic Blacks (NH Blacks) have had persistently higher levels of disability compared to non-Hispanic Whites (NH Whites), while differences between Hispanics and NH Whites have been more variable [32–35]. Moreover, racial and ethnic minorities have historically experienced higher disability rates and faced greater exposure to economic shocks from the 2008 recession [27] There is a lack of evidence at the national level on whether age-specific racial/ethnic disparities in disability have been changing over the last 20 years [36] examined trends in reported functional limitations by race/ethnicity in U.S. adults between 1999 and 2018. They found increases in reported limitations across all racial/ethnic groups (Asians, Blacks, Latino/Hispanic, and White) with Latino/Hispanics experiencing the largest increase over the period. Mahajan et al.'s (2021) analyses combined adults of all ages (18 + years). The study did not analyze how trends may differ across age groups (e.g., working vs. retirement age) [23] found an increasing trend in ADL limitation prevalence between 2000 and 2014 for the population aged 65+ in all educational levels (except for those with education beyond high school), but did not study racial disparities.. There has not been a follow up yet comparing those trends before and after the great recession, nor any evidence indicating their race and ethnicity composition.

This study examines age-specific disability trends among adults aged 50−84 by sex, age group, and race/ethnicity in the United States in the period prior to the COVID-19 pandemic from 2000 to 2018. Understanding trends in disability in the pre-pandemic period provides essential baseline information on the underlying health of the population prior to the disruption of the pandemic and can be used as a gauge to identify how the pandemic may have shifted long-term trends in disability. Furthermore, different definitions of disability need to be studied separately because they might not all present the same trajectories across time [12].

We had two primary objectives. The first was to investigate whether age-specific disability trends changed between the decades of 2000–2009 and 2010–2018, considering this possible shift in trajectories across decades. The second was to investigate changes over time in racial/ethnic disparities in disability over these decades.

## Methods

We used data from the National Health Interview Survey (NHIS), a large national in-person survey that provides population-based estimates of health in the United States. With weighting, each NHIS survey is nationally representative of the non-institutionalized U.S. population of the 50 states and the District of Columbia. NHIS data have been harmonized in the Integrated Public Use Microdata Series (IPUMS) data library [37], which is free to use and download and was used in this analysis after registration (which can be found in https://nhis.ipums.org/nhis/). We examined disability trends from 2000 to 2018 by sex (female, male) and three age groups (50–64, 65–74, and 75–84). We use the following terms to refer to the age groups: middle-age (50–64), young-old (65–74), and old (75–84). We excluded the 2019 survey because the measurement of limitations in the NHIS changed, making it non-comparable with previous years for these indicators.

Public-use files of the NHIS top-code age at 85 years, hence we excluded those 85 years and over as we did not have information on how the age distribution of this group changed over time and we were unable to control for single years beyond that age in our models. We originally restricted the analyses to three racial/ethnic groups (non-Hispanic whites, non-Hispanic blacks, Hispanics) because other racial/ethnic groups had sample sizes insufficient to study trends. However, since 40% of the Hispanic population is reported to being Foreign-born, we decided to stratify nativity trends separately between the Hispanics who were US-Born and Foreign-born, resulting in four groups. S1 Table in the supplementary material indicates the share of Foreign-born individuals by race/ethnicity. Unlike Hispanics, The share of Foreign-Born Non-Hispanic Whites and Blacks was less than 10%,. This grouping meant that nativity (U.S. born-Foreign-born), originally a covariate in our models, was removed from the final analysis in non-Hispanic groups due to the small-sample size (and as a result any change in prevalence trends in Foreign-born non-Hispanic groups might be

due to randomness rather than an epidemiological pattern). In addition, since the literature has acknowledged the healthy migrant effect across Hispanics, we believe it warrants separate analysis of trends for this group [32,34,38].

The presence of difficulties in performing tasks of daily life independently has traditionally been used to monitor disability levels and trends in population studies [39]. We separately analyzed limitations in Activities of Daily Living (ADLs) and Instrumental Activities of Daily Living (IADLs). Limitations in ADLs include impairments in basic tasks. We defined an ADL limitation as a reported limitation in at least one of the following activities: toileting, eating, bathing, dressing, and transferring from bed to chair. Limitations in IADLs emphasize the interaction between functional ability and social or economic tasks. We defined an IADL limitation as a reported limitation in at least of one of the routine needs such as everyday household chores, doing necessary business, shopping, or getting around for other purposes, because of a physical, mental, or emotional problem. The NHIS asked respondents a single question about whether they had any of these difficulties and recorded the response as a binary outcome, so the activities were not addressed individually. S6 Table in the supplementary material provides the codes for the variables analyzed.

Sampling weights provided by NHIS were used at the three levels: primary sampling unit, strata (to have a more precise variance estimation) and individual person weights. R 4.3.1 statistical software was used for our analyses. The multi-layered survey design was addressed in the analysis by using the *survey* package.

We fit survey-weighted logistic regression models predicting ADL and IADL limitations, stratified by sex and age group (50–64, 65–74, 75–84 years), yielding a total of 12 models. Each model included a variable for Year (centered at 2010), Age (continuous), Race-Ethnicity (4 groups – non Hispanic-Whites [NHWs], non-Hispanic-Blacks [NHBs], US-born Hispanics [UBHs] and Foreign-born Hispanics [FBHs]) and Educational attainment (3 categories: no high school diploma, high school diploma/GED, bachelor's degree or higher ([BA or beyond]). An interaction term for post-2010 trends was added in all models to observe variations in trends across the two decades. We also considered interactions between time (years) and the four race/ethnicity groups.

From these models, we generated predicted probabilities of disability for each race/ethnicity group at three time points (2000, 2010, 2018), holding age and education constant at reference values. For our primary inferences about temporal changes, we focus on confidence intervals for predicted prevalence and risk ratios rather than individual model coefficients (these are available in the S2 Table in the supplementary material), as these provide more interpretable estimates of substantive change. The level of statistical significance was p < .05 for all hypothesis tests. We calculated absolute changes in prevalence (percentage points), percent changes, and risk ratios comparing prevalence across time periods. 95% confidence intervals for changes and risk ratios were computed using the log-transforming of risk ratios, accounting for uncertainty in model predictions. Goodness of fit R-square tests are provided in S3 Table in the supplementary material. To account for multiple testing and avoid false positives across the 180 model coefficients, we applied false discovery rate (FDR) correction using the Benjamini-Hochberg procedure [40], available in S4 Table in the supplementary material. To account for multicollinearity, we estimated the Variance Inflation Factor (VIF), in S5 Table in the supplementary material.

The overall analytical sample size was 478,868 between 2000 and 2018, with 477,171 complete cases. We removed 1,697 cases that were missing the Foreign-born variable, which represented the 0.35% of the sample (details are provided in S1 Table in the supplementary material). We opted to use complete-case analysis since we believe our estimates would not be affected by such scale.

Sensitivity analyses presented in the supplementary material include age-specific predicted probabilities of the prevalence trends analysis of the without sampling (S4-S6 Figs), models without controlling by education (S7-S9 Figs), and replications of the analyses with difficulties in walking without especial equipment as a proxy for ADL and having any memory troubles to do activities as a proxy for IADL, available in S10 to S12 Figs). Additionally, we conducted post-hoc power analyses [41] using two-sample tests of proportions (significance at p < .05) to assess the model's ability to detect the observed changes in disability prevalence between decades across all sex, age, and race/ethnicity subgroups (S6 Table).

An Open Science Framework repository with the corresponding R code was created to replicate the analyses: https://osf.io/6pfth/?view_only=f460587c0f91438b872812b403d51e49).

## Results

Table 1 presents descriptive statistics by race/ethnicity, decade, and age group. Across all racial/ethnic groups, the prevalence of ADL limitations increased modestly between 2000–2009 and 2010–2018, with the most notable increases among those ages 50–64. Among NHWs in this age group, ADL prevalence increased from 1.51% to 1.93%, while IADL prevalence increased from 3.42% to 3.95%. NHBs exhibited higher baseline disability levels, with ADL prevalence increasing from 3.26% to 3.86% and IADL prevalence from 6.53% to 6.99% in the 50–64 age group.

Hispanic populations showed distinct patterns by nativity. Among UBHs aged 50–64, ADL prevalence increased from 2.73% to 3.75% and IADL prevalence from 5.63% to 6.61%. In contrast, FBHs in this age group maintained the lowest disability levels of all groups, though with slight increases (ADL: 1.50% to 1.58%; IADL: 2.57% to 2.82%).

**Table 1. Descriptive characteristics by decade and age group, ages 50-84. Mean values and Standard Deviations in parentheses.**

| Age Group | Period | Observations | Females (%) | Age (mean, years) | ADL Limitation (%) | IADL Limitation (%) | BA or Higher (%) |
|---|---|---|---|---|---|---|---|
| **NH-White** | | | | | | | |
| 50-64 | 2000-2009 | 96,347 | 51.14 (0.16) | 56.26 (0.01) | 1.51 (0.04) | 3.42 (0.06) | 33.00 (0.15) |
| 50-64 | 2010-2018 | 109,240 | 51.5 (0.15) | 56.88 (0.01) | 1.93 (0.04) | 3.95 (0.06) | 34.56 (0.14) |
| 65-74 | 2000-2009 | 38,764 | 53.35 (0.25) | 69.19 (0.01) | 2.85 (0.08) | 5.83 (0.12) | 24.23 (0.22) |
| 65-74 | 2010-2018 | 52,115 | 52.25 (0.22) | 69.00 (0.01) | 2.99 (0.07) | 5.50 (0.1) | 33.05 (0.21) |
| 75-84 | 2000-2009 | 26,397 | 58.52 (0.30) | 78.95 (0.02) | 6.42 (0.15) | 13.61 (0.21) | 19.82 (0.25) |
| 75-84 | 2010-2018 | 27,677 | 55.29 (0.30) | 78.93 (0.02) | 6.43 (0.15) | 12.09 (0.20) | 24.94 (0.26) |
| **NH-Black** | | | | | | | |
| 50-64 | 2000-2009 | 18,738 | 56.23 (0.36) | 55.95 (0.03) | 3.26 (0.13) | 6.53 (0.18) | 20.24 (0.29) |
| 50-64 | 2010-2018 | 21,806 | 55.86 (0.34) | 56.52 (0.03) | 3.86 (0.13) | 6.99 (0.17) | 21.49 (0.28) |
| 65-74 | 2000-2009 | 6,655 | 58.63 (0.60) | 69.02 (0.03) | 5.95 (0.29) | 11.10 (0.39) | 15.72 (0.45) |
| 65-74 | 2010-2018 | 7,980 | 56.47 (0.56) | 68.88 (0.03) | 6.50 (0.28) | 10.56 (0.34) | 19.19 (0.44) |
| 75-84 | 2000-2009 | 3,520 | 65.26 (0.80) | 78.83 (0.05) | 12.27 (0.55) | 23.38 (0.71) | 14.94 (0.60) |
| 75-84 | 2010-2018 | 3,757 | 62.82 (0.79) | 78.73 (0.05) | 13.18 (0.55) | 20.50 (0.66) | 17.09 (0.61) |
| **Hispanic US-Born** | | | | | | | |
| 50-64 | 2000-2009 | 8,064 | 55.52 (0.55) | 55.92 (0.05) | 2.73 (0.18) | 5.63 (0.26) | 14.05 (0.39) |
| 50-64 | 2010-2018 | 8,473 | 54.76 (0.54) | 56.26 (0.05) | 3.75 (0.21) | 6.61 (0.27) | 18.42 (0.42) |
| 65-74 | 2000-2009 | 2,941 | 55.42 (0.92) | 69.16 (0.05) | 5.58 (0.42) | 9.96 (0.55) | 9.32 (0.54) |
| 65-74 | 2010-2018 | 2,964 | 54.99 (0.91) | 68.85 (0.05) | 5.74 (0.43) | 10.56 (0.56) | 14.98 (0.66) |
| 75-84 | 2000-2009 | 1,521 | 59.24 (1.26) | 78.68 (0.07) | 12.1 (0.84) | 20.91 (1.04) | 8.74 (0.72) |
| 75-84 | 2010-2018 | 1,439 | 56.78 (1.31) | 78.84 (0.07) | 12.02 (0.86) | 18.28 (1.02) | 11.26 (0.83) |
| **Hispanic Foreign-Born** | | | | | | | |
| 50-64 | 2000-2009 | 12,665 | 52.59 (0.44) | 55.73 (0.04) | 1.5 (0.11) | 2.57 (0.14) | 14.14 (0.31) |
| 50-64 | 2010-2018 | 14,468 | 51.88 (0.42) | 55.90 (0.04) | 1.58 (0.10) | 2.82 (0.14) | 14.94 (0.30) |
| 65-74 | 2000-2009 | 3,852 | 56.78 (0.80) | 68.88 (0.05) | 4.31 (0.33) | 7.09 (0.41) | 13.53 (0.55) |
| 65-74 | 2010-2018 | 4,028 | 56.80 (0.78) | 68.82 (0.05) | 4.79 (0.34) | 7.89 (0.42) | 13.75 (0.54) |
| 75-84 | 2000-2009 | 1,868 | 60.12 (1.13) | 78.61 (0.06) | 11.51 (0.74) | 17.24 (0.87) | 11.99 (0.75) |
| 75-84 | 2010-2018 | 1,892 | 59.73 (1.13) | 78.75 (0.06) | 13.58 (0.79) | 18.60 (0.89) | 13.00 (0.77) |

Note: Data are from the National Health Interview Survey, 2000–2018. Estimates reflect sample, unit and strata weighting.

Among those aged 65–74, disability levels were relatively stable across decades for most groups, though UBHs showed modest increases in both ADL (5.58% to 5.74%) and IADL (9.96% to 10.56%) limitations. In the oldest age group (75–84), ADL prevalence remained stable or increased slightly across all racial/ethnic groups, with FBHs women showing a notable increase from 11.51% to 13.58%. IADL prevalence in this age group generally declined, particularly among NHWs (13.61% to 12.09%) and NHBs (23.38% to 20.50%).

Compositional changes in the population over the decades were modest, though each age group and racial/ethnic category showed a growing percentage of individuals with a bachelor's degree or higher. The most dramatic educational gains occurred among NHWs aged 65–74 (from 24.23% to 33.05% with Bachelor's Degree and Beyond) and the oldest age group across all racial/ethnic categories.

Figs 1 and 2 present the predicted probabilities for the years 2000, 2010 and 2018. Figs 3, 4, 5 present the relative risk ratio estimates in the respective age groups (50 –64, 65–74, 75–84) estimated from the regression models for the years 2010 compared to 2000, 2018 compared to 2010, and finally, 2018 compared to 2000. Figs 6, 7, 8 present the change in prevalence for the same period respectively. Model coefficients are available in the Supplementary Material (S2 Table) along with figures highlighting the predicted probability by year for the whole period of analysis (S1 to S3 Figs in the supplementary material). FDR testing (S4 Table in the supplementary material) found only one coefficient with a corrected p-value higher than 0.05: the interaction term between trends after the year 2010 and the FBH group in the model predicting the IADL prevalence for Males aged 65–74 (Model 8). In any case, the model does not produce a significant change across the risk ratios as Fig 2 indicates so this is not relevant.

Across all age groups in 2000, NHB men and women had considerably higher levels of both disability measures compared to their NH White counterparts. Hispanics presented a more complex picture: UBHs generally had disability levels

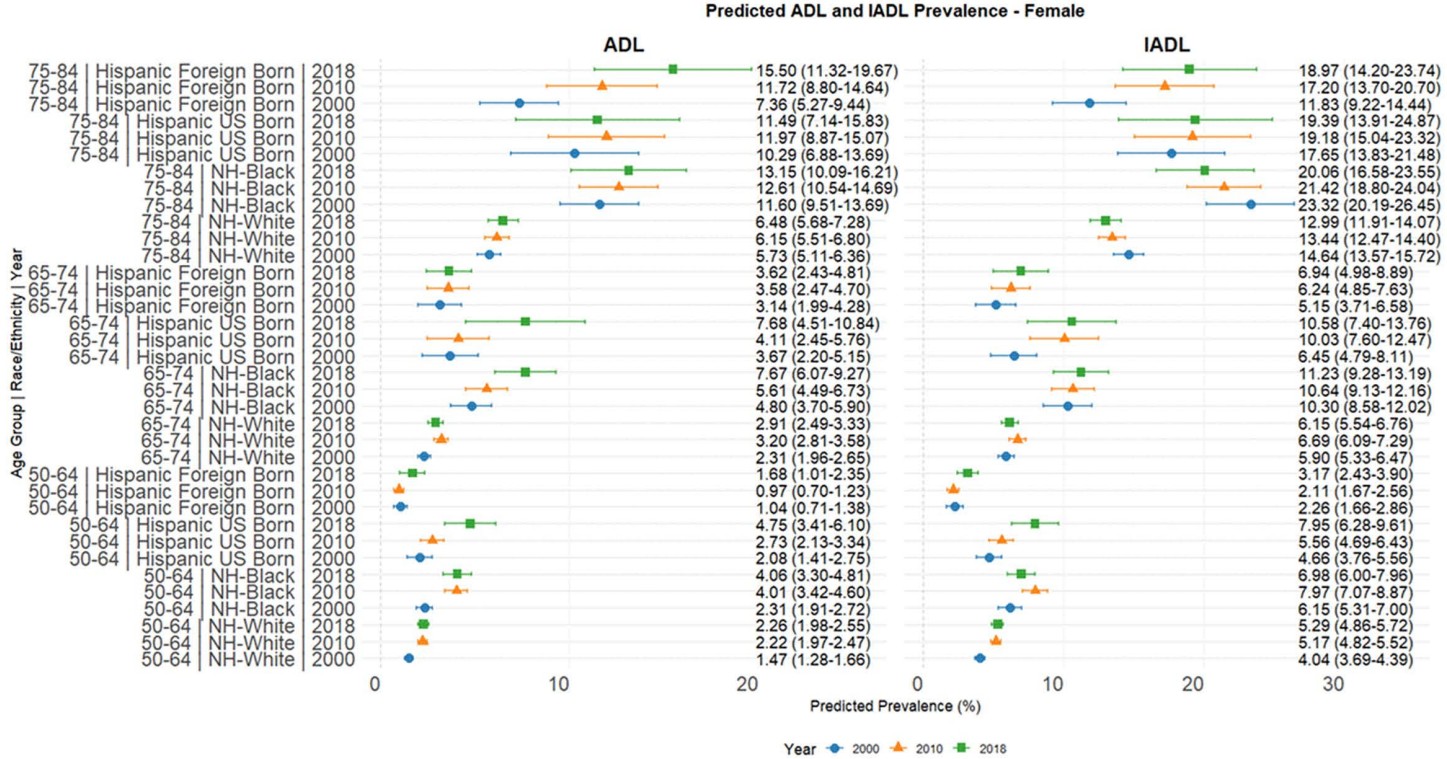

**Fig 1. Predicted ADL and IADL prevalence by age group in years 2000, 2010 and 2018 based on models inS2 Table, Females.**

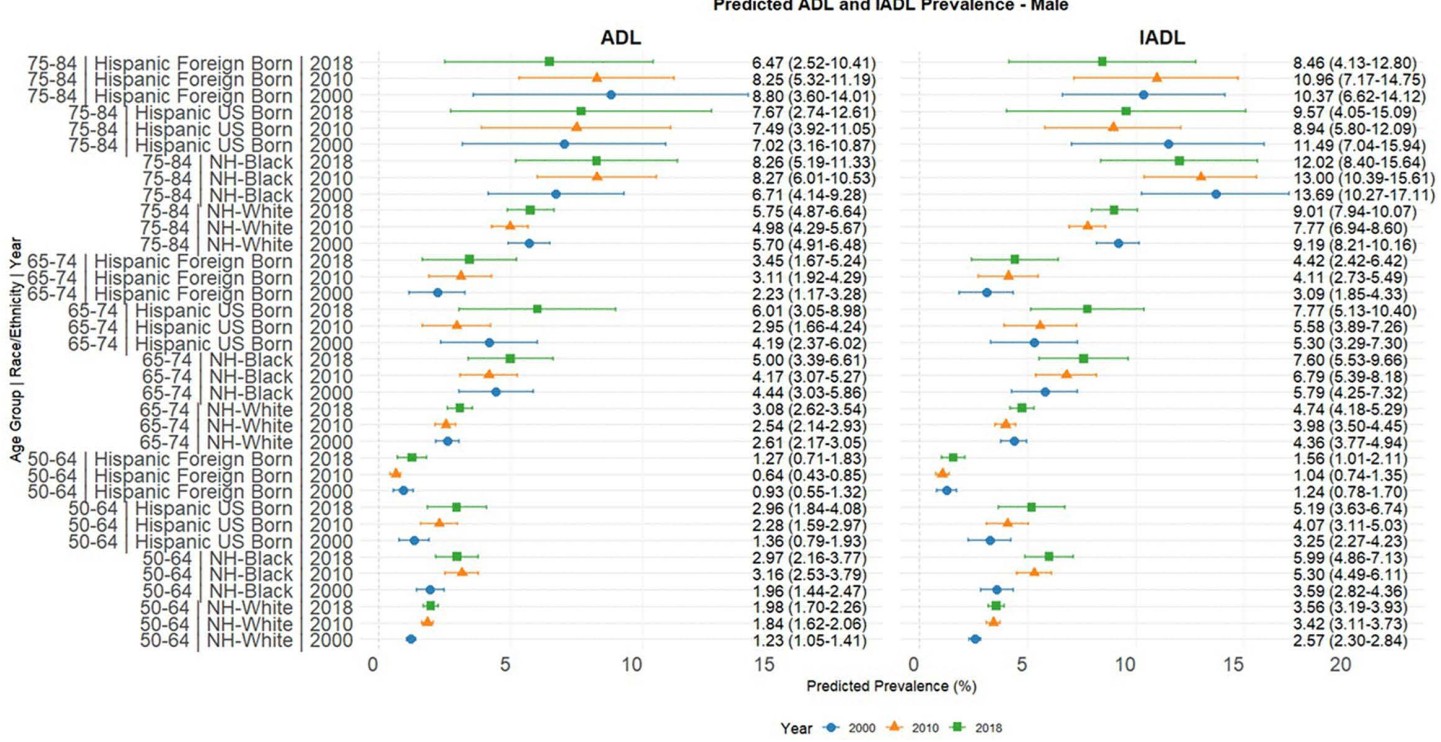

**Fig 2. Predicted ADL and IADL prevalence by age group in years 2000, 2010 and 2018 based on models in S2 Table, Males.**

comparable to or higher than NH Whites, while FHBs typically had the lowest baseline disability prevalence of all groups. These patterns evolved differently across the two decades, with particularly notable changes among UBHs and FHBs Foreign-born Hispanic women in older age groups.

Our estimates in Figs 2 and 3 confirm the higher level of disability among women compared to men in all racial/ethnic groups and at all ages, consistent with previous literature [22,42] We did not find strong evidence of a change in the gender gap across decades.

The group of middle-aged adults (ages 50–64) experienced the strongest changes in disability prevalence in relative terms, with the largest relative risk increases observed in this age group. Between 2000 and 2010, disability levels increased substantially for NHWs and NHBs. NHW females experienced a 51% (Risk Ratio = 1.51, 95% CI: 1.27–1.79) and 28% (RR = 1.28, 95% CI: 1.15–1.43) increase in the risk of ADL and IADL limitations, respectively, while males presented similar increases. NHB men and women, who had the highest prevalence in 2000, experienced even larger relative increases (62% for ADLs among men [RR = 1.62, 95% CI: 1.16–2.26] and 74% among females [RR = 1.74, 95% CI: 1.38–2.19]). During this first decade, both UBHs and FBHs experienced modest increases, with FBHs maintaining the lowest disability levels of all groups.

Trends shifted markedly in the second decade (2010–2018). NHWs and NHBs experienced minimal changes, with both groups plateauing at their elevated 2010 levels. In contrast, Hispanics experienced substantial increases during this period, particularly UBHs. Among UBH women, ADL limitations increased by 74% (RR = 1.74, 95% CI: 1.21–2.49) and IADL limitations by 43% (RR = 1.43, 95% CI: 1.10–1.86) between 2010 and 2018, resulting in a disability prevalence spike of 2.02 points (CI = 0.55–3.49) for ADL and 2.39 points (CI = 0.51–4.27) for IADL, approaching prevalence levels of NHBs. FBHs also experienced significant increases in the second decade (74% for ADLs [RR = 1.74, 95% CI: 1.07–2.82] and

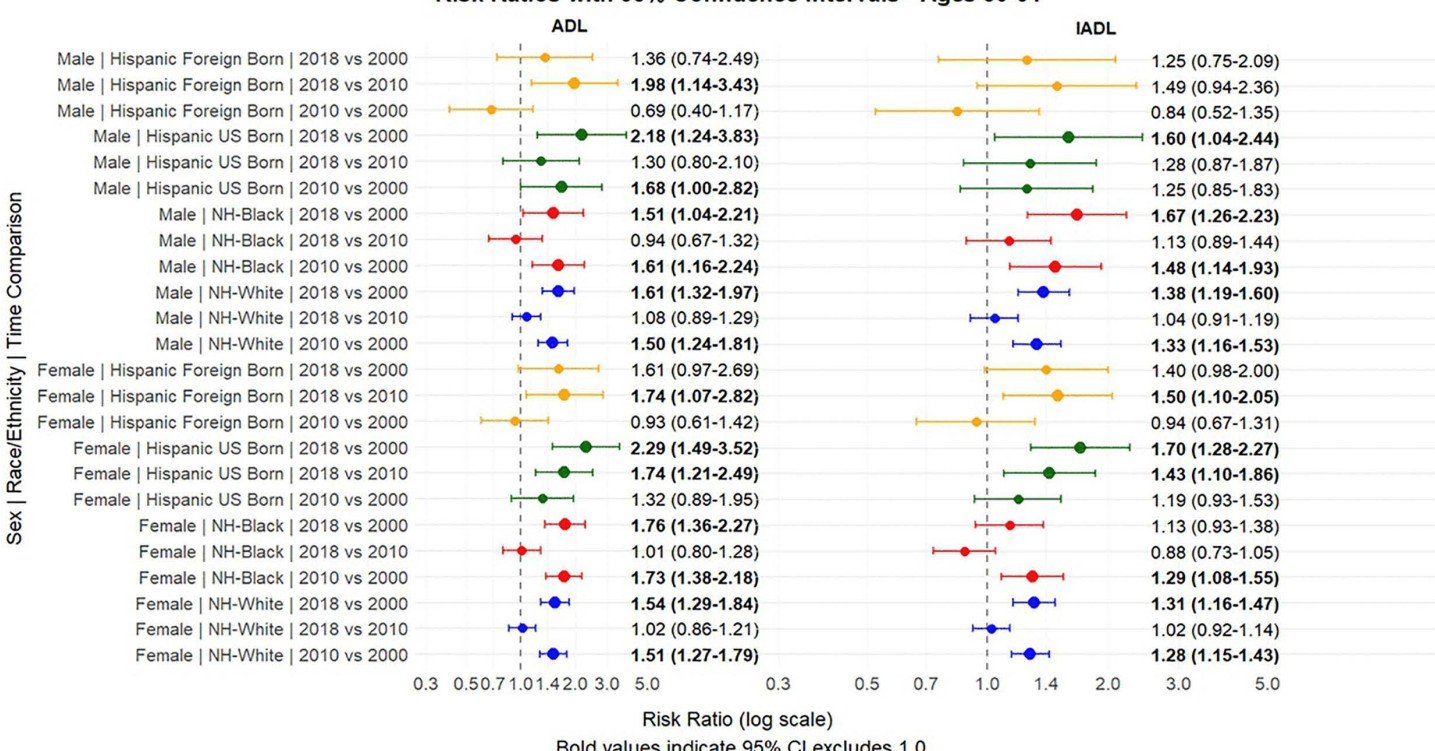

**Fig 3. Risk Ratios between 2000-2010 and 2010-2018 and 2000-2008, with 95% Confidence Intervals by sex based on predicted probabilities from Models in** <u>S2 Table</u>**, Ages 50-64.**

50% for IADLs [RR = 1.50, 95% CI: 1.10–2.05] among women), though their absolute prevalence levels remained lower than other groups and changes in prevalence might be minimal due to confidence intervals. Over the entire 2000–2018 period, UBH women experienced a 129% increase in ADL limitations (RR = 2.29, 95% CI: 1.49–3.52) and a 71% increase in IADL limitations (RR = 1.71, 95% CI: 1.28–2.27), which was the largest relative increases of any subgroup in this age range, driven mostly by increases in the 2010–2018 period.

We must acknowledge that while this age group presented the largest changes in relative risk, absolute prevalence levels remained lower than in older age groups, as <u>Figs 1</u>, <u>2</u>, <u>3</u> (and Figs 1A–3A in the supplementary material) indicate.

Young-old adults (ages 65–74) also showed a pattern of increasing disability levels over the 2000–2018 period in certain groups, though changes were generally smaller than in the 50–64 age group. NHB women experienced a 59% increase in ADL limitations (RR = 1.59, 95% CI: 1.16–2.17, p < .05) over the period. Among Hispanics, US-born individuals experienced particularly notable increases: ADL limitations more than doubled among UBH women (109% increase, RR = 2.09, 95% CI: 1.17–3.71) over the entire period, rising from 3.69% in 2000 to 7.69% in 2018, while IADL limitations increased by 64% (RR = 1.64, 95% CI: 1.10–2.44). While NHWs also presented increases in ADL limitations among women, these increases substantially widened the gap between them and UBHs. FBHs in this age group experienced more modest increases, with changes not reaching statistical significance.

Older adults (ages 75–84) displayed more heterogeneous patterns, with opposing trends for ADLs and IADLs. For ADL limitations, the most striking finding was among FBH women, who experienced a 110% increase over the entire period (RR = 2.10, 95% CI: 1.42–3.11, with a prevalence change of 8.14% reaching a value of 15.56%), representing the largest

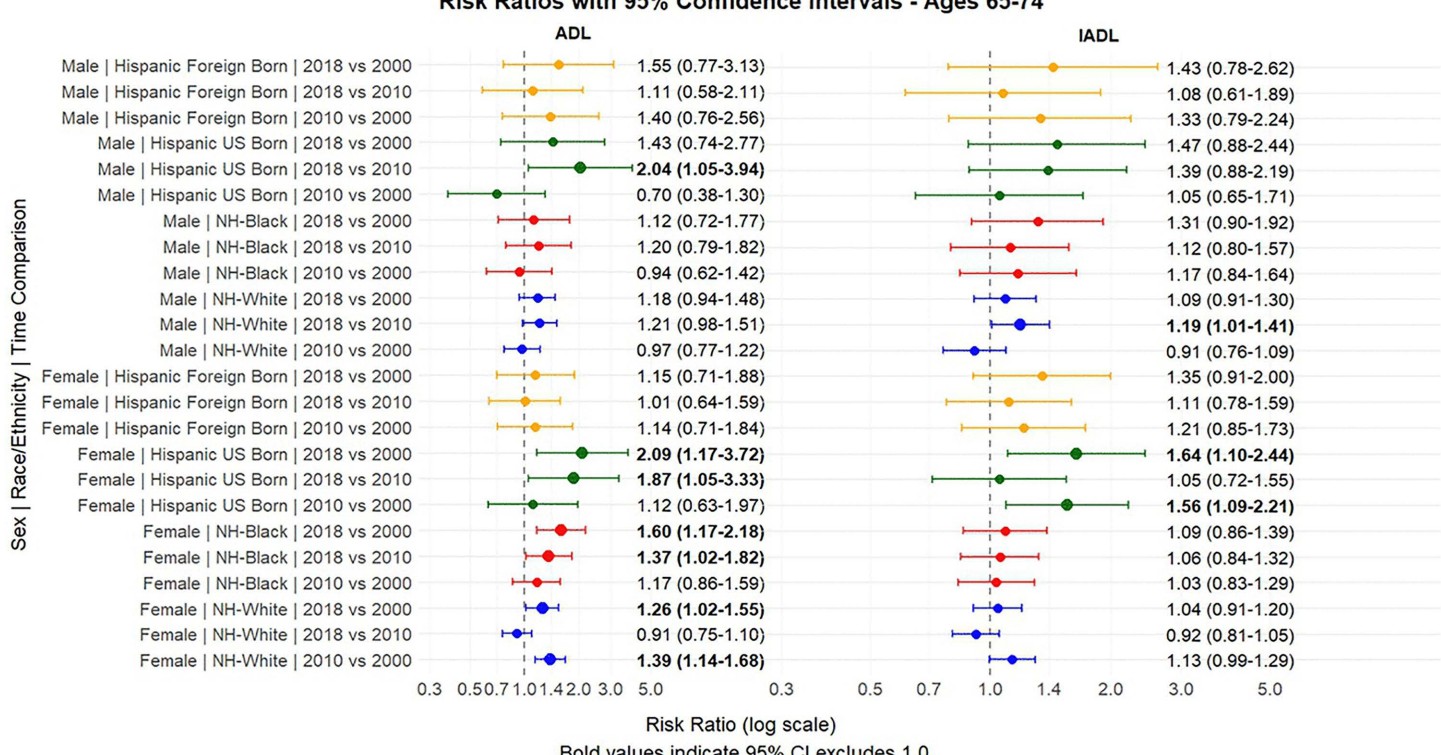

**Fig 4. Risk Ratios between 2000-2010 and 2010-2018 and 2000-2008, with 95% Confidence Intervals by sex based on predicted probabilities from Models in S2 Table, Ages 65-74.**

absolute increase in this age group, surpassing all other demographic groups by 2018. UBH women also showed elevated levels but with more modest increases. For IADL limitations, NHW men and women experienced modest declines (11% decline among women, RR = 0.89, 95% CI: 0.79–0.99), while FBH women showed a 60% increase (RR = 1.60, 95% CI: 1.15–2.24) over the period. UBHs had relatively stable IADL levels in this age group.

## Discussion

Disability levels among U.S. adults declined substantially over the 1980s and 1990s [13]. Earlier research has shown that these declines ended in the first decade of the 2000s [14,20]. There have been few studies that have examined disability trends since 2010 and prior to the COVID-19 pandemic, a period when U.S. life expectancy trends were stagnant. The few existing studies have not considered trends within age groups, critical for the assessment of cohort-specific dynamics in health and impacts on families and the economy. Our analysis reveals three main findings. First, there was no evidence that disability levels improved in the decade prior to the pandemic. Thus, the general adverse trends recorded in the first decade of the 2000s,while stagnated in most groups, did not show a decline into the 2010–2018 period. Second, there is considerable heterogeneity in post-2000 disability trends by age group. Among the age groups studied, middle-aged adults (ages 50–64) displayed the most adverse trends with considerable increases in both ADL and IADL limitations. Much of the increase is attributable to rising disability in the 2000–2010 period for NHWs and NHBs, but Hispanics were the most affected group in the 2010–2018 period, with this pattern being particularly evident among the middle-aged group (ages 50–64). Third, when disaggregating the Hispanic population between those who were born in the United

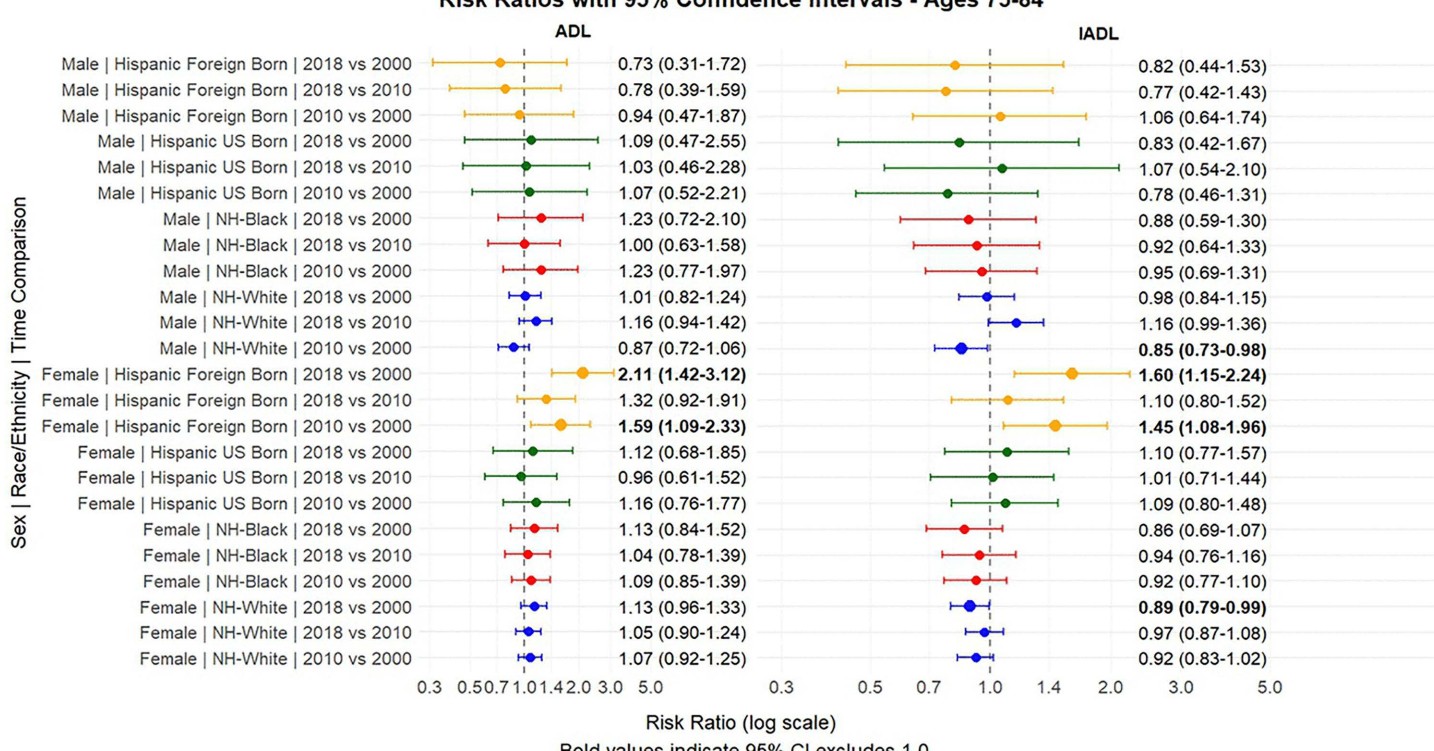

**Fig 5. Risk Ratios between 2000-2010 and 2010-2018 and 2000-2008, with 95% Confidence Intervals by sex based on predicted probabilities from Models in S2 Table, Ages 75-84.**

States and those who did not, we find divergent patterns: UBHs experienced the steepest relative increases among middle-aged adults, while FBHs—despite starting with the lowest disability levels—showed substantial increases particularly among older women. Disability levels were relatively stagnant among the young-old (ages 65–74), although there was a tendency toward increase in some cases. Patterns among the old (ages 75–84) indicated stagnation, but with some heterogeneity in the trend across the two disability measures, with the exception of FBH women.

Our findings align with prior research documenting rising disability prevalence among Americans across a range of pre-retirement ages. For instance, [12] reported increasing rates of ADL and IADL limitations among adults aged 18–44 over the same period, suggesting a broader trend of worsening functional health in younger cohorts. Similarly, [43] found that between 1992 and 2014, pain prevalence increased more rapidly among adults aged 55–71 than among those aged 72 and older, with the steepest increases observed among men between 2000 and 2010, followed by a period of stagnation. Another study [31] showed that the pain prevalence gap between adults aged 45–64 and those aged 65–84 narrowed around 2010 but widened again by 2018, driven by stagnating pain levels in the younger group and increases among older adults. In our analysis, we observed a similar post-2010 increase only in specific subgroups—namely, Hispanics in the 50–64 age group, NHBs and UBHs aged 65–74, and FBHs aged 75–84—whereas other groups showed more stable patterns over the same period.

Two main challenges are present in studying national disability trends. The first is understanding the true implications of the rising trends in the prevalence of traditional disability measures such as ADLs and IADLs among those under age 75 remain relatively low in absolute terms. For example, the prevalence of ADL limitations among those ages 50–64 were

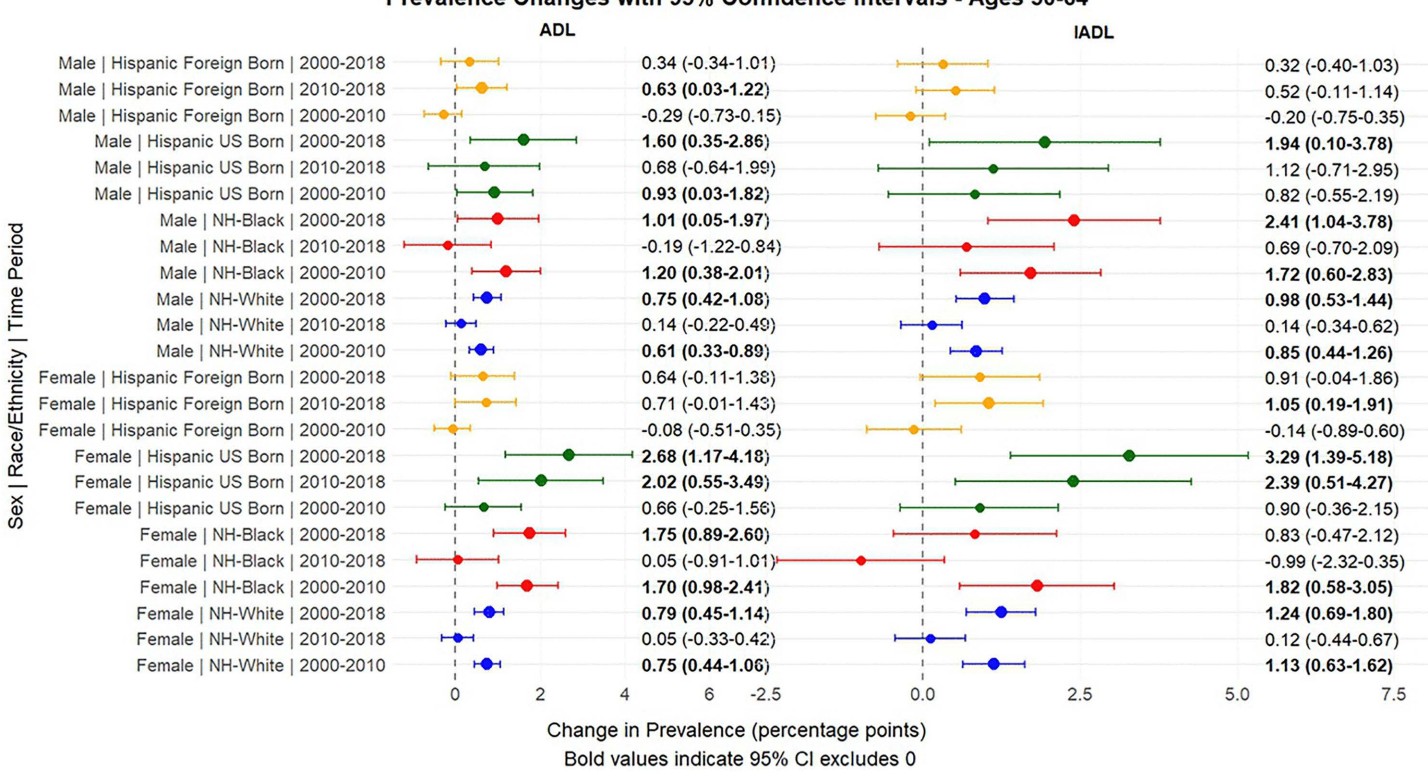

**Fig 6. Prevalence change between 2000-2010, 2010-2018, 2000-2018, with 95% Confidence Intervals by sex based on predicted probabilities from Models in S2 Table, Ages 65-74, Ages 50-64.**

in the range of 2–6%, so while the lack of positive trends is a sign of concern, is still is the age group that presents better overall population health than older age groups. However, the implications of rising disability in this pre-retirement age group extend beyond individual health to workforce participation, economic productivity, and family caregiving burdens. Also, many of the changes we observed over time after mandatory retirement ages did not reach statistical significance despite using samples from the largest national in-person survey amenable to the study of long-run trends. This was particularly true for some Hispanic subgroups, where the wide confidence intervals reflect smaller sample sizes. Despite lacking statistical significance in some cases, the magnitude of increases we observed among Hispanics -particularly the doubling of ADL limitations among UBH women aged 65–74 and the 110% increase among FBH women aged 75–84 represent substantively important changes that warrant attention and suggest the importance for oversampling Hispanic population in future surveys. Unlike mortality, which has a clear definition based on a single event that is undeniable (dying), disability encompasses a gamut of different health states. Disability can be operationalized through activities of daily living or functional limitations or other domains and can be assessed either through respondent-reports or clinical measurement. As a result, evidence from trends analyses may differ depending on the measure of disability, particularly when definitions embrace different dimensions of well-being and health. This ambiguity can be partially due to the subjective nature of assessments, but also because these definitions may represent different dimensions of impairment, especially given changing technologies and environments that mitigate functional difficulties [44]. We examined two standard measures of disability, the existance of difficulties to carry ADLs and IADLs in one survey. The field would benefit from future efforts examining a wider set of disability measures (especially when it comes to non-physical activities, similar to

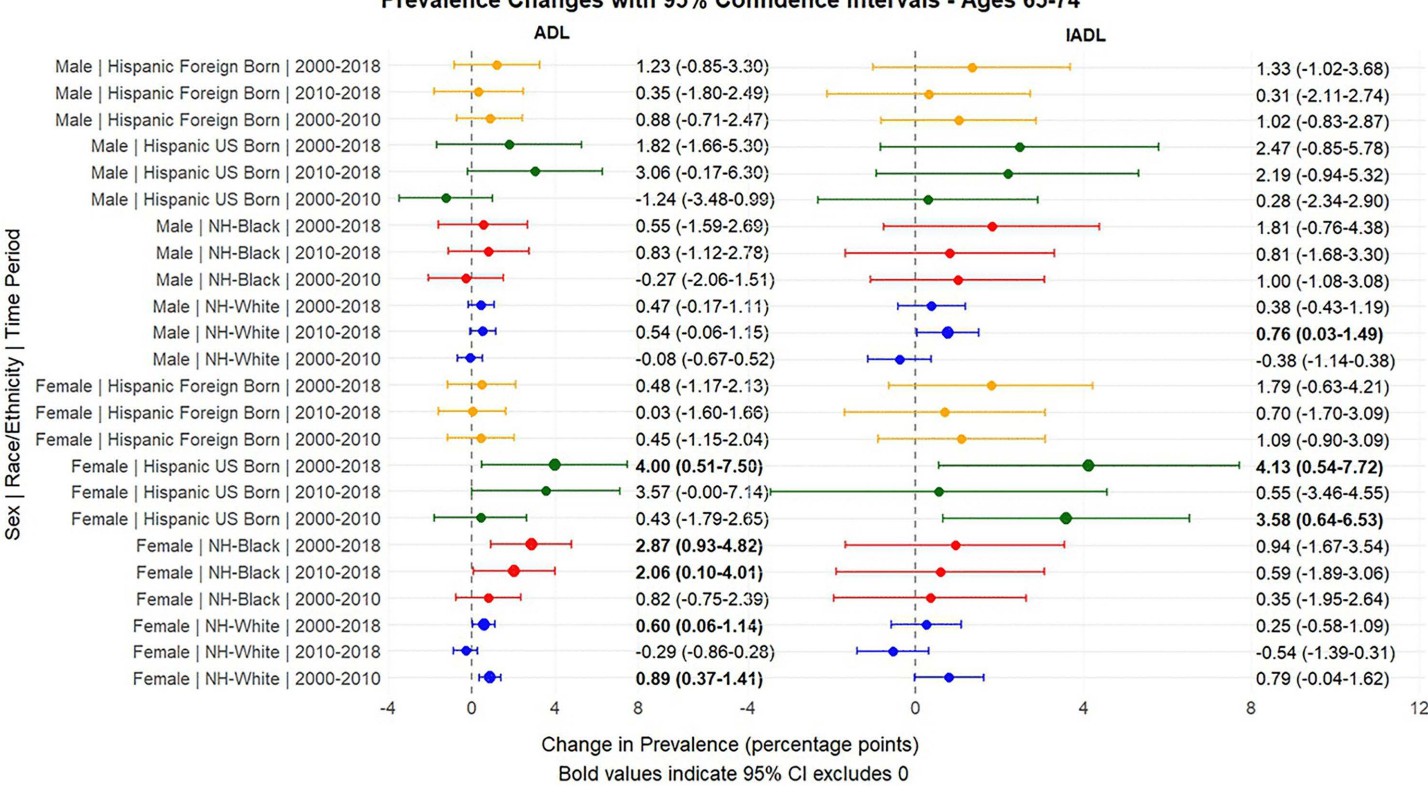

**Fig 7. Prevalence change between 2000-2010, 2010-2018, 2000-2018, with 95% Confidence Intervals by sex based on predicted probabilities from Models in** S2 Table**, Ages 65-74, Ages 65-74.**

the ones compressing the IADL indicator, that are somewhat lacking in the survey), and, importantly, consistencies across national datasets, as has been done in earlier studies of U.S. disability trends [13].

The continued increase in disability among middle-aged through 2010–2018 for Hispanics in particular (independently of if they were born in the United States or not) and the stagnation observed for other groups is alarming as it represents nearly two decades of deteriorating health in this population. We did not examine causes of the trends, but multiple distal and proximal causes are likely at play. The general deterioration of the broader social and economic circumstances of middle-aged Americans has received considerable attention. Embedded within this context have been stark rises in obesity and related cardio-metabolic risk factors that have co-occurred with rising disability, the opioid epidemic, and increasing prevalence of certain cancers in younger Americans [45]. Moreover, we have lost the benefit of steep rises in educational attainment that was characteristic of middle-aged adults in earlier periods. The temporal divergence in trends, with Non-Hispanic Whites and Blacks experiencing increases primarily in 2000–2010 followed by stagnation, while Hispanics showed acceleration in 2010–2018 warrants further investigation. This pattern might reflect differential impacts of the Great Recession and subsequent recovery, changes in immigration patterns and policies affecting the Hispanic population's composition and economic and health security, or different timing in the downstream health effects of rising obesity and cardiometabolic conditions across racial/ethnic groups.

With respect to racial/ethnic disparities, we found no evidence that the large disparities existent in 2000 between NHWs and NHBs, and to some extent, between NHWs and Hispanics were mitigated over the nearly two decades examined. Rather, disparities appeared to increase in some cases and trends that originally favored Hispanics were even reversed.

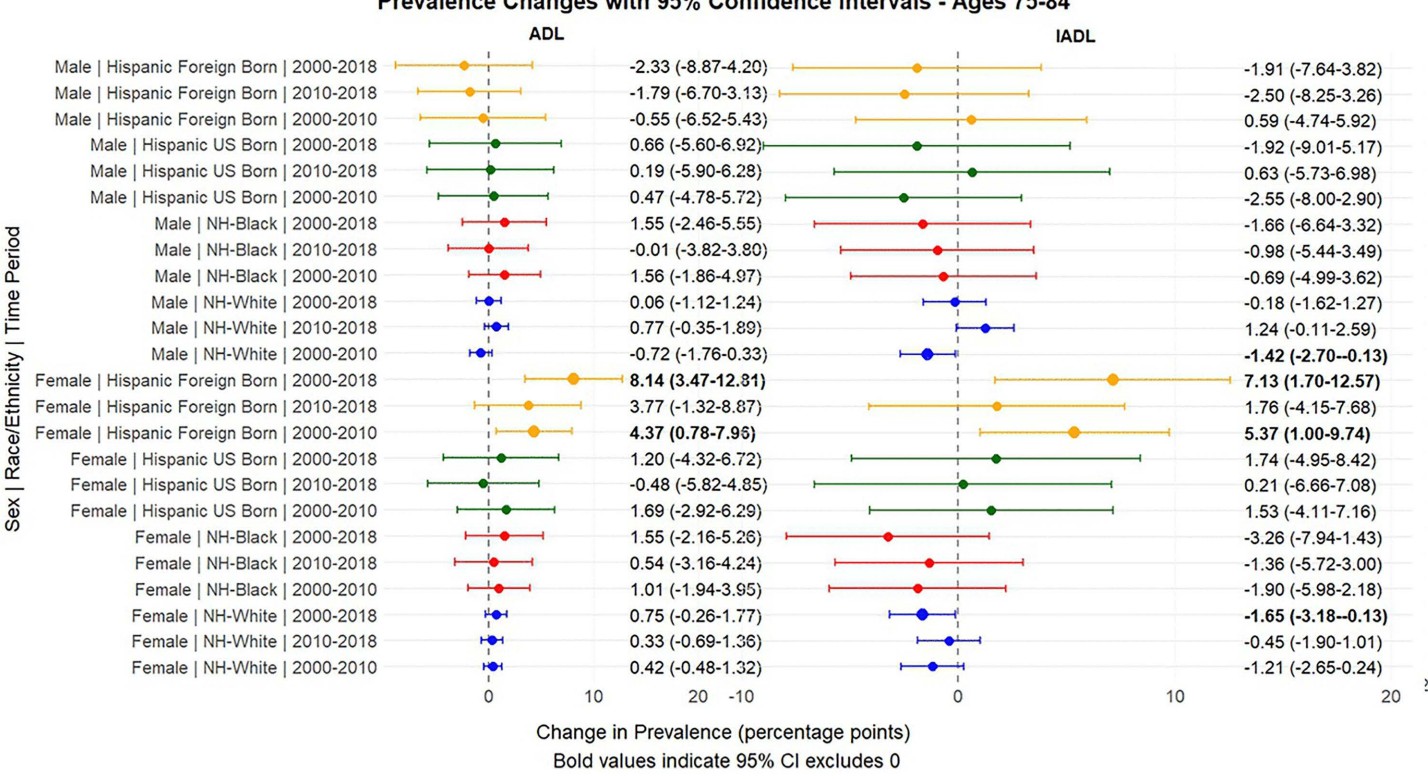

**Fig 8. Prevalence change between 2000-2010, 2010-2018, 2000-2018, with 95% Confidence Intervals by sex based on predicted probabilities from Models in S2 Table, Ages 65-74Ages 75-84.**

For example, among the middle-aged (50 –64), NHW and NHB men had a similar relative increase in ADL risk between 2000 and 2018, but NHBs had a 73% higher baseline risk in 2000, implying that levels in disability in this population in absolute terms grew faster compared to NHWs. For IADLs, the relative growth for NHB men was higher than in NHW men. A similar pattern emerged for women and in some cases in the older age groups, with absolute growth in disability levels in NHBs outpacing that of NHWs. In any case, these patterns of increase indicate no dissipation of prevalence gaps between the rest of the groups and the NHWs, despite their own prevalence increase.

The steep increase in disability among Hispanics is a key and concerning conclusion from our study. A critical finding is the divergent patterns between US-born and Foreign-born Hispanics. Foreign-born Hispanics exhibited the lowest disability levels in 2000 across all age groups, consistent with the healthy immigrant effect. However, this advantage eroded substantially over time, particularly among older Foreign-born Hispanic women (ages 75–84), who experienced a 110% increase in ADL limitations between 2000 and 2018 (RR = 2.10, 95% CI: 1.42–3.11), resulting in the highest disability prevalence of any demographic group by 2018 (15.56%). This pattern suggests that the protective effects associated with immigrant status diminish with duration of U.S. residence and/or age, potentially due to cumulative exposure to adverse labor conditions, limited access to healthcare, acculturative stress, and adoption of less healthy behaviors.

In contrast, UBHs demonstrated disability levels comparable to or higher than NH Whites at baseline and had the steepest relative increases among middle-aged adults (ages 50–64). US-born Hispanic women in this age group experienced a 129% increase in ADL limitations over the study period (RR = 2.29, 95% CI: 1.49–3.52), the largest relative increase of any subgroup. This suggests that younger cohorts of UBHs may be particularly vulnerable to the deteriorating

health conditions affecting middle-aged Americans more broadly, possibly compounded by other socioeconomic factors that differentially affect this population, even after controlling for education in our models.

Understanding trends in this population requires a recognition of the highly dynamic composition of this population given high levels of immigration, and to some extent emigration, over this period. Over time, there will be increases in the number of Hispanics aged 50 + who have spent a considerable portion of their lives in the United States with an increasing number being born in the country. Our disaggregation by place of birth (citizenship status is a restricted variable in the IPUMS-NHIS files, so we make no distinction of that status) reveals that both US-born and Foreign-born Hispanics experienced steep increases, but with distinct age patterns and likely different causal pathways. UBHs may be affected by the same factors driving disability increases among middle-aged Americans generally (obesity, opioid use, economic precarity), but compounded by structural disadvantages including discrimination, occupational segregation into lower-quality jobs and barriers to healthcare access. FBHs may face cumulative effects of migration-related stressors, concentration in physically demanding occupations and limited access to preventive healthcare, and the erosion of protective practices over time. While we can only postulate, steep rises in obesity and diabetes in this population may play a critical role as to are adverse labor force exposures (e.g., physically demanding and dangerous jobs) and exposures to structural features of U.S. society (e.g., racism and socio-economic vulnerabilities).

The question of whether disability trends continued to worsen, stabilize, or improve after 2010 has important implications for multiple domains. From a policy perspective, rising disability among working-age adults affects labor force participation and Social Security Disability Insurance demand, while increasing disability at older ages places pressure on Medicare, Medicaid, and family caregivers. From a health equity standpoint, the factors driving increases in midlife mortality have disproportionately affected certain demographic groups, yet racial and ethnic minorities have historically faced higher disability rates and greater economic vulnerability. Understanding whether the adverse trends of the 2000s persisted into the 2010s—and whether they affected all demographic groups equally—is essential for projecting future healthcare needs, identifying populations most at risk, and informing targeted interventions to improve population health and reduce disparities.

These trends have implications for social programs, healthcare systems, and economic outcomes. Rising disability among middle-aged adults (ages 50–64) is particularly concerning as this population is in their prime working years. Increased disability in this age group likely contributes to declining labor force participation rates, may increase demand for Social Security Disability Insurance, and places caregiving burdens on families. The substantial increases among Hispanics are especially concerning given this population's rapid growth and their overrepresentation in physically demanding occupations without adequate workplace protections or health insurance coverage. Healthcare systems and social service providers should anticipate increased demand for disability-related services, particularly from younger cohorts entering older ages with higher disability burdens than previous generations. Moreover, these findings suggest that disparities in disability may widen further without targeted interventions addressing the social determinants of health and structural barriers faced by minoritized populations.

This study has several limitations that should be considered when interpreting our findings. First, the NHIS relies on self-reported data, which may be subject to reporting bias and influenced by respondents' understanding of disability questions, cultural interpretations of functional limitations, and language barriers, particularly among foreign-born populations. Second, our analysis is descriptive in nature and does not employ a causal framework; therefore, we cannot identify the mechanisms driving the observed trends or establish causal relationships between demographic characteristics and disability changes. Third, our parsimonious modeling approach intentionally focused on demographic covariates (age, race/ethnicity, education and time period) to clearly describe population-level trends, but this means we did not account for other potentially important factors such as wealth, chronic disease burden, occupational exposures, or healthcare access that may contribute to or confound the observed patterns. Fourth, sample sizes were limited for certain demographic subgroups, and we could not consider other ethnic groups. Despite these limitations, the large, nationally representative

sample and consistent data collection methods across nearly two decades provide valuable insights into evolving disability patterns across diverse demographic groups in the United States.

In conclusion, this study addresses gaps in our understanding of national disability trends by age and race/ethnicity in the United States. We examined trends prior to the onset of the COVID-19 pandemic, and these results can serve as an evidence base to understand how the pandemic may (or may not) have changed longer-term trends in disability in the U.S. population. Given that the populations experiencing the steepest increases in disability—middle-aged adults, Hispanics, and NH Blacks—were also disproportionately affected by COVID-19, the pandemic may have accelerated these adverse trends, making continued monitoring even more critical We document several long-term adverse trends in disability that require continued monitoring and explanations, **as well as policy interventions to address widening disparities and support aging populations entering later life with greater disability burdens than previous cohorts**.

## Supporting information

**S1 Table. Foreign-born and US-born observations among complete cases by race-ethnicity.**
(DOCX)

**S2 Table. Logistic regression results for all 12 models, stratified by age and sex.**
(DOCX)

**S3 Table. Goodness of fit R-square tests for all 12 models.**
(DOCX)

**S4 Table. Coefficients that changed p-value after False Discovery Rate (FDR) corrections using Benjamini-Hochberg procedure.**
(DOCX)

**S5 Table. Maximum Variance Inflation Factor (VIF) testing for multicollinearity in models 1–12.**
(DOCX)

**S6 Table. Post-hoc power analyses of proportions across groups by sex, age, and race-ethnicity.**
(DOCX)

**S7 Table. Names and values for key variables in the analyses.**
(DOCX)

**S1 Fig. Predicted probabilities for ADL and IADL limitation prevalence by sex and race/ethnicity, ages 50–64.**
(TIF)

**S2 Fig. Predicted probabilities for ADL and IADL limitation prevalence by sex and race/ethnicity, ages 65–74.**
(TIF)

**S3 Fig. Predicted probabilities for ADL and IADL limitation prevalence by sex and race/ethnicity, ages 75–84.**
(TIF)

**S4 Fig. Predicted probabilities for difficulty walking and memory problems by sex and race/ethnicity, ages 50–64.**
(TIF)

**S5 Fig. Predicted probabilities for difficulty walking and memory problems by sex and race/ethnicity, ages 65–74.**
(TIF)

**S6 Fig. Predicted probabilities for difficulty walking and memory problems by sex and race/ethnicity, ages 75–84.**
(TIF)

**S7 Fig. Predicted probabilities for ADL and IADL not controlled by education, ages 50–64.**
(TIF)

**S8 Fig. Predicted probabilities for ADL and IADL not controlled by education, ages 65–74.**
(TIF)

**S9 Fig. Predicted probabilities for ADL and IADL not controlled by education, ages 75–84.**
(TIF)

**S10 Fig. Unweighted predicted probabilities for ADL and IADL, ages 50–64.**
(TIF)

**S11 Fig Unweighted predicted probabilities for ADL and IADL, ages 65–74.**
(TIF)

**S12 Fig. Unweighted predicted probabilities for ADL and IADL, ages 75–84.**
(TIF)

## Author contributions

**Conceptualization:** Octavio Bramajo, Moumita Chakraborty, Neil K. Mehta.

**Data curation:** Octavio Bramajo, Brandon O'Grady.

**Formal analysis:** Octavio Bramajo, Brandon O'Grady, Moumita Chakraborty, Neil K. Mehta.

**Funding acquisition:** Neil K. Mehta.

**Investigation:** Octavio Bramajo, Brandon O'Grady, Moumita Chakraborty, Neil K. Mehta.

**Methodology:** Octavio Bramajo, Moumita Chakraborty, Neil K. Mehta.

**Project administration:** Octavio Bramajo, Neil K. Mehta.

**Resources:** Octavio Bramajo, Neil K. Mehta.

**Software:** Octavio Bramajo, Neil K. Mehta.

**Supervision:** Moumita Chakraborty, Neil K. Mehta.

**Validation:** Octavio Bramajo, Brandon O'Grady, Moumita Chakraborty, Neil K. Mehta.

**Visualization:** Octavio Bramajo.

**Writing – original draft:** Octavio Bramajo, Moumita Chakraborty, Neil K. Mehta.

**Writing – review & editing:** Octavio Bramajo, Moumita Chakraborty, Neil K. Mehta.

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
