## [Decision Letter · Decision Letter 0]

9 Sep 2025

Dear Dr. Bramajo,

Thank you for submitting your manuscript to PLOS ONE. After careful consideration, we feel that it has merit but does not fully meet PLOS ONE’s publication criteria as it currently stands. Therefore, we invite you to submit a revised version of the manuscript that addresses the points raised during the review process.

We look forward to receiving your revised manuscript.

Kind regards,

Ryota Sakurai, Ph.D.

Academic Editor

PLOS ONE

Journal Requirements:

Additional Editor Comments:

Two reviewers made some comments on this paper. Please respond to all these comments.

Reviewers' comments:

Reviewer's Responses to Questions

**Comments to the Author**

1. Is the manuscript technically sound, and do the data support the conclusions?

Reviewer #1: No

Reviewer #2: Yes

2. Has the statistical analysis been performed appropriately and rigorously?

Reviewer #1: No

Reviewer #2: Yes

3. Have the authors made all data underlying the findings in their manuscript fully available?

Reviewer #1: Yes

Reviewer #2: No

4. Is the manuscript presented in an intelligible fashion and written in standard English?

Reviewer #1: No

Reviewer #2: Yes

Reviewer #1: Interesting paper however there is major deficits regarding the robustness explained in the statsitifcal analysis, clarity in the paper and flow of the language. Minor grammatical and phrasing issues

Results portrayal should be improved

Reviewer #2: The manuscript addresses an important topic using a large, nationally representative dataset (NHIS/IPUMS). The overall study design and analytical framework are appropriate, and the findings are broadly consistent with prior research on stagnating disability declines and widening disparities. However, several methodological and reporting issues need to be addressed before the manuscript is technically sound:

Survey design adjustments – It is unclear whether strata and PSU design features were fully accounted for. This is essential to obtain correct standard errors.

Trend specification – The use of a single linear breakpoint at 2010 requires stronger justification and sensitivity checks (e.g., splines, joinpoint, 2008/2014 breakpoints).

Covariates – Education may act as a mediator; sensitivity analyses without education are recommended. Nativity × race/ethnicity interactions should be explored given compositional changes.

ADL/IADL definitions – A supplementary mapping table of items and a stricter outcome definition (≥2 limitations) would strengthen transparency and robustness.

Supplementary material errors – Several tables contain inconsistencies (coefficients vs. SEs vs. p-values). These must be corrected and regenerated directly from code.

Reporting – Both relative risks and absolute prevalence differences should be presented. Multiple testing should be acknowledged (FDR or exploratory framing).

Presentation – The manuscript is intelligible and written in standard English, though some sentences require minor copy-editing for clarity. Terminology should be standardized across text and tables.

Data and code availability – Authors should provide the IPUMS extract ID and a public repository for analysis code to comply with PLOS ONE data policies.

**Do you want your identity to be public for this peer review?** For information about this choice, including consent withdrawal, please see our Privacy Policy

Reviewer #1: No

Reviewer #2: No

---

## [Author Response · Author response to Decision Letter 1]

10 Nov 2025

We thank each reviewer for their careful assessment of our manuscript and for their suggested improvements. Below, we outline the changes we have made to the made to the manuscript.

Reviewer 1:

Introduction

Comment

Recommend not starting the first sentence with “After”

Response

We have edited the first sentence accordingly.

Comment

Recommend rephrasing the line “another important story”

Response

We have edited the sentence accordingly.

Comment

The reviewer provided several comments around defining disability and limitations:

Disability has been used numerous times during the introduction however this was not defined clearly.

Needs earlier definition as most of the understanding surrounding the introduction revolves on the reader having a clear idea and concept of disability.

The reader does not have clear definitions and understandings of disability and limitations therefore following along the ideas that are being presented remains ambiguous. It would be helpful to put in hard factual numerical data to guide the reader as to the current and or past understanding regarding disability to give a frame of reference.

Response

We added the following paragraphs in the introduction section, expanding on the disability measures, concepts and previous studies.

“Limitations in activities of daily living (ADLs) and instrumental activities of daily living (IADLs) have long been considered gold standard measures for assessing functional disability in population-based research (Katz, 1963, 1983; Lawton & Brody, 1969). These measures capture fundamental aspects of independent functioning—from basic self-care tasks like bathing and dressing (ADLs) to more complex activities like managing finances and medications (IADLs). These measures are often used in the context of the disablement process (Verbrugge & Jette, 1994), a framework that demonstrates how chronic conditions, injuries, and impairments lead to functional limitations that hinder quality of life and the ability to perform daily activities (Verbrugge et al., 1989; Zajacova & Margolis, 2025). Over the 1980s and 1990s, the U.S. experienced sizeable declines in national-level disability prevalence as measured by ADLs and IADLs (Freedman et al., 2002; Martin et al., 2010; Schoeni et al., 2001; Waidmann & Liu, 2000). Various explanations for this improvement have been identified including increases in educational attainment (Freedman et al., 2002; Martin et al., 2010; Schoeni et al., 2001), improvements in medical care, and the long-term benefits of improvements in earlier-in-life health (Freedman & Martin, 1998). The first decade of the 2000s brought about a notable shift in U.S. disability trends measured by these indicators. Previous studies that analyzed this period found that the declining ADL and IADL prevalence among Americans aged 55 and older stagnated (Freedman et al., 2013; Lakdawalla et al., 2004; Seeman et al., 2010; Tsai, 2017). Some other studies suggest that disability increased among middle-aged individuals, reaching by 2010 a 1.7% prevalence of ADL and 4% of IADL for those aged between 40 and 65, and above near 9% and 12% respectively for individuals aged 65+ (Freedman et al., 2013; Martin, Freedman, et al., 2010; Martin & Schoeni, 2014). Other group classifications presented similar trends: among those aged 55-69, IADL limitations increased by an average of 1.33% annually between 1998-2012, with most of the increase concentrated toward the end of the period (Choi et al., 2016). “

Comment

The majority of the introduction remains speculation on previous studies and pointing out unclear aspects of literature however without context the importance of these discrepancies have minimal impact on the reader

The overall pace and macrostructure of the introduction remains vague and does not guide the reader through specific points adequately and easily

Response

We have re-structured the Introduction to better motivate the analysis. As noted above, we provide a more detailed and careful review of the existing literature on national disability trends. We then include a dedicated paragraph on why studying disability trends in the post-2010 period is important, starting in page 2 with this paragraph:

Trends after 2010 remain understudied. Around 2010, U.S. life expectancy began to stagnate and even decline driven by stagnating declines in cardiovascular disease mortality and rising mortality from drug overdoses, suicides, and alcoholic liver disease. Broader deteriorations in population health have also been documented after 2010 (Case & Deaton, 2015, 2017). The 2008 economic crisis and its aftermath provide additional motivation for examining the post-2010 period. Economic recessions have been shown to have lasting effects on population health, particularly through increased financial strain, job loss, reduced access to healthcare, and heightened psychological distress (Burgard & Kalousova, 2015; Catalano et al., 2011). It remains unclear whether this event is also associated with a turning point for disability trends. These effects may be particularly pronounced for middle-aged adults approaching retirement who experienced job displacement during peak earning years. Moreover, the post-2010 period saw accelerating increases in obesity levels and the continued escalation of the opioid epidemic, which are key risk factors for functional limitations (Alley & Chang, 2007; Stokes et al., 2020; Zajacova et al., 2021).

Statistical Analysis

1. No power calculation shown.

Response:

It's unclear which kind of power analysis is requested here, but to comply with this suggestion, we conducted post-hoc power analyses (Table 6S in the supplementary material) to assess our ability to detect changes in disability prevalence between the two decades (2000-2009 vs 2010-2018) using two-sample tests of proportions (significance lower than 0.05). Our primary findings regarding disability trends among middle-aged adults (50-64) were well-powered for Non-Hispanic Whites, Non-Hispanic Blacks and US-Born Hispanics. Statistical power was more limited for older age groups (65-74 and 75-84) and smaller demographic subgroups.

However, we note that power analyses are more commonly applied to hypothesis-testing studies designed to detect specific effect sizes, rather than to descriptive epidemiological studies documenting population-level trends as this one. Our study's primary objective is to describe and characterize temporal patterns in disability prevalence across demographic groups, not to test a priori hypotheses about specific differences. The substantial sample sizes from the NHIS provide sufficient precision to detect meaningful population-level changes, as evidenced by the narrow confidence intervals around our prevalence estimates. Moreover, similar descriptive trend analyses in the disability literature cited in the paper (i.e: Choi et al., 2016; Freedman et al., 2013; Martin, Freedman, et al., 2010) have not required formal power calculations, as the goal is comprehensive description rather than hypothesis confirmation.

2. What calculation was done to ensure the model was well fitted vs overfitted ????

Response:

We now provide model fit estimates based on R-squared values (Supplementary Material, Table S3). We computed three types of R-squared (McFadden, Nagelkerke, and Pseudo R-squared), which all yielded similar values ranging from 0.018 to 0.047 across models. While these values appear modest, we believe they are appropriate and expected for this type of analysis. Our models intentionally include only key demographic covariates (age, race/ethnicity, education, and time period) because our goal is to describe population-level epidemiological trends rather than to build predictive models or explain individual-level variation in disability. In large population-based studies examining broad temporal patterns, low R-squared values are typical and do not indicate poor model performance. The precision of our estimates is reflected in the narrow confidence intervals rather than in R-squared values. Furthermore, our parsimonious modeling approach focuses on main effects and key interactions guards against overfitting. The consistency of findings across different model specifications and the social plausibility of the observed trends provide additional confidence in our results.

3. No sensitivity analysis was noted

Response:

We now provide several sensitivity analyses, which are presented in the supplementary material, from figures 4S to 12S, analyzing the age-specific predicted probabilities of the following:

-Models without survey weights to illustrate the effect of the complex survey design. The results remained consistent with our main findings.

-We tested models without education to assess its influence. Results were largely similar across groups, with the exception of the Hispanic foreign-born subgroup, which showed some variation.

-Due to limitations in the data (i.e., individual ADL/IADL items were not asked separately but altogether in a single question), we used proxy measures. Specifically, we used difficulty walking without equipment as a stand-in for ADL limitations, and presence of memory problems (if the respondent had activities limited by difficulty remembering things) as a proxy for IADL limitations. While these are imperfect substitutes, they capture related functional impairments. We acknowledge that stricter definitions (e.g., ≥2 limitations) would be ideal but the questions asked are not exactly the ones that make the ADL/IADL definitions, and we are open to exploring additional proxies available in the dataset if the reviewer deems it necessary.

Across all sensitivity analyses, predicted probabilities remained consistent with our main estimates, supporting the robustness of our findings.

4. No correction for multiple comparisons was mentioned e.g Bonferroni

Response:

We used False Discovery Rate (FDR) adjustment, using the Benjamini-Hochberg procedure, available in Table 4S of the supplementary material. Only one term that originally had a p value lower than 0.05 exceeded that threshold, but it does not imply a significant trend change either way.

5. How was missing data managed

Response:

We changed the methods section and described which observations were removed in order to work with a complete case-analysis. Results are Observations on missing data are Table 1S of the supplementary material

6. No notation for test of multicollinearity

Response:

We now are showing tests for multicollinearity in Table 5S of the supplementary material. In all cases the Variance Inflation Factor (VIF) results were under 3 (which is under the rule of thumb threshold value of 5 to consider that there might be multicollinearity among variables)

7. No external validation or bootstrapping mentioned

Response:

We did not perform external validation as our primary goal was to describe trends at a population level using standard epidemiological techniques, not building a predictive model of disability. Our methods and results seem to be consistent with related studies that we mention in the discussion.

8. Data on the software information needs to be listed

Response:

We clarified now the software in the methos section. We used R software, version 4.3.1 in the methods sections and specially the survey package, that we used for our estimations.

Overview: Deficit noted regarding the robustness of the statistical analysis to cement the clarity of the information.

Response:

Thank you for your suggestions We agree with the reviewer and we changed our analysis accordingly as shown.

Results

What is BA? This acronym has not been defined however is utilized in the Tables 2. Table 2, 3 and 4 needs better formatting.

Recommend dividing table and having the 95% confidence interval as a separate column. Very busy and over populated

3. Would be helpful to have the information compared within ethnicity along the age groups to see a trend among the ethnicity in one format eg. Graph or Bar chart

Overview: This data can be better represented in formats that highlight trends especially in the distinct groups.

Table format can still be utilized however visual format of the data would help increase the strength of the information represented Discussion

Response:

Thank you for your suggestions. We agree with the reviewer. This was clarified (Bachelor’s Degree and Beyond). Also, we redid the tables and presented figures instead (Figures 1 and 2) because we believe is easier to understand, but showing the same results as the tables before.

1. Helpful to add the factual data in the 1980s and 1990s to help place the current numerical data in context from this study

2. Grammatical error for “ is still is the age group…” in 2nd paragraph

3. Helpful to add some data on the factors that have “ co - occurred with rising disability”

Response:

We modified the grammar and added a mention of a Freedman et al. (2004) study quantifying the declining prevalence trend in the 1990 for ADL and IADL among Americans.

4. How can you explain the with the NH Black population

Response:

There might be something missing here with the reviewer’s comment, so we were unable to address this.

5. Limitations of the study is missing

Response:

We added the following section in the discussion, near the end of the manuscript:

This study has several limitations that should be considered when interpreting our findings. First, the NHIS relies on self-reported data, which may be subject to reporting bias and influenced by respondents' understanding of disability questions, cultural interpretations of functional limitations, and language barriers, particularly among foreign-born populations. Second, our analysis is descriptive in nature and does not employ a causal framework; therefore, we cannot identify the mechanisms driving the observed trends or establish causal relationships between demographic characteristics and disability changes. Third, our parsimonious modeling approach intentionally focused on demographic covariates (age, race/ethnicity, education and time period) to clearly describe population-level trends, but this means we did not account for other potentially important factors such as wealth, chronic disease burden, occupational exposures, or healthcare access that may contribute to or confound the observed patterns. Fourth, sample sizes were limited for certain demographic subgroups, and we could not consider other ethnic groups. Despite these limitations, the large, nationally representative sample and consistent data collection methods across nearly two decades provide valuable insights into evolving disability patterns across diverse demographic groups in the United States.

6. Further exploration and utilization of the data is needed to be mentioned as well

7. How can this data be helpful from a clinical perspective? Recommendation - Major Revision

Response:

This is a comment that makes us think a bit more about our findings, thanks. We believe the most important thing about this paper are the substantial increases in disability among middle-aged adults (ages 50-64) have important clinical implications. This pre-retirement age group experienced the largest relative increases in functional limitations, with some subgroups—particularly US-born Hispanic women—experiencing larger increases in ADL limitations between 2000 and 2018. From a clinical perspective, these findings highlight the need for earlier screening and preventive interventions targeting functional decline in patients in their 50s and early 60s, and this is the trend among all race/ethnicity groups, but with changes in intensity. Additionally, clinicians and geriartricians should be aware that the traditional focus on disability management in adults aged 65+ may need to shift earlier in the life course, as functional limitations are increasingly manifesting during the working years when individuals may still be employed and lack access to Medicare. Early identification and intervention during middle age, when functional decline may be more modifiable, could help prevent or delay more severe disability in later life and

---

## [Decision Letter · Decision Letter 1]

26 Dec 2025

Age-specific Trends in Limitations of Daily Activities in American Adults aged 50-84 by Race and Ethnicity, 2000-2018

PONE-D-25-30426R1

Dear Dr. Bramajo,

We’re pleased to inform you that your manuscript has been judged scientifically suitable for publication and will be formally accepted for publication once it meets all outstanding technical requirements.

Kind regards,

Ryota Sakurai, Ph.D.

Academic Editor

PLOS One

Additional Editor Comments (optional):

The manuscript improved with comments from the reviewers. Thanks for providing important information.

Reviewers' comments:

Reviewer's Responses to Questions

**Comments to the Author**

Reviewer #2: All comments have been addressed

Reviewer #3: All comments have been addressed

2. Is the manuscript technically sound, and do the data support the conclusions?

Reviewer #2: Yes

Reviewer #3: Yes

3. Has the statistical analysis been performed appropriately and rigorously?

Reviewer #2: Yes

Reviewer #3: Yes

4. Have the authors made all data underlying the findings in their manuscript fully available?

Reviewer #2: Yes

Reviewer #3: Yes

5. Is the manuscript presented in an intelligible fashion and written in standard English?

Reviewer #2: Yes

Reviewer #3: Yes

Reviewer #2: Thank you for the opportunity to review your manuscript. The study addresses an important public health topic and uses a valuable long-term data source. The research question is well motivated and the findings are relevant for understanding trends and disparities in functional limitations among older adults in the United States.

Reviewer #3: The authors have carefully and adequately addressed all of the reviewer’s comments, and the manuscript has been substantially improved.

**Do you want your identity to be public for this peer review?** For information about this choice, including consent withdrawal, please see our Privacy Policy

Reviewer #2: No

Reviewer #3: No

---

## [Editor Report · Acceptance letter]

PONE-D-25-30426R1

PLOS One

Dear Dr. Bramajo,

I'm pleased to inform you that your manuscript has been deemed suitable for publication in PLOS One. Congratulations! Your manuscript is now being handed over to our production team.

Kind regards,

on behalf of

Dr. Ryota Sakurai

Academic Editor

PLOS One